# The Digital Mind: How Computers (Re)Structure Human Consciousness

**Brian L. Ott**

Department of Communication, Missouri State University, Springfield, MO 65897, USA;
brianott@missouristate.edu

**Abstract:** Technologies of communication condition human sense-making. They do so by creating the social environment we inhabit and extending their structural biases and logics through human use. As such, this essay inquires into the prevailing habits of mind in the digital era. Employing a media ecology of communication, I argue that digital computers and microprocessors are defined by three structural properties and, hence, underlying logics: digitization (binary code), algorithmic execution (input/output), and efficiency (machine logic). Repeated exposure to these logics cultivates a digital mind, a model of thinking, communicating, and sense-making characterized by intransigence, impertinence, and impulsivity. I conclude the essay by exploring the broader implications of a digital mind, paying particular attention to the challenges it poses to democratic politics.

**Keywords:** digital culture; media ecology; habits of mind; digital computing; microprocessors; binary code; closed system; machine logic





## 1. Introduction

Over the course of the past two decades, our world has steadily become more digital. Today, we inhabit a social, political, and cultural environment that is so dominated by digital technologies that the extraordinary impact and influence of those technologies on all aspects of our lives is virtually impossible to see, let alone to comprehend. The seeds of this transformation, of this paradigm shift, in daily life and, even more fundamentally, in what it means to be human, can be traced back to the invention, development, and spread of the microprocessor. Outside of the high-tech fields of information and computer science, we typically do not hear much about microprocessors today and even less about how they matter and what their ubiquitous presence and operation is doing to us and our culture. But, like all technologies, microprocessors are neither neutral nor inert. They are engineered to function in particular ways and, as such, have a series of inherent or built-in structural biases [1]. Repeated exposure to those biases alters how we communicate and interact, how we engage with and respond to events, and even how we make sense of ourselves and our world. In short, our prevailing technologies of communication endlessly and fundamentally remake us in their image.

More than a decade ago, Gary Small and Gigi Vorgan observed:

> The current explosion of digital technology not only is changing the way we live and communicate but is rapidly and profoundly altering our brains. Daily exposure to high technology—computers, smart phones, video games, search engines [...]—stimulates brain cell alteration and neurotransmitter release, gradually strengthening new neural pathways in our brains while weakening old ones. Because of the current technological revolution, our brains are *evolving* right now—at a speed like never before [2] (p. 1).

While the present essay is not directly concerned with neuroscience and how digital media alter the human brain [3], it is interested in how the digital revolution conditions our habits of mind and our modes of information processing and sense-making [1]. My central aim in

this essay, adopting the perspective of media ecology, is to identify and explain the key features of what I term the *digital mind*. The digital mind is a shorthand designation for the ways digital computing generally and microprocessing technology specifically have altered not so much *what* we think but *how* we think and, subsequently, how we act. As such, I seek to map the character of consciousness—the prevailing patterns of thought—in a digital culture [2]. But, first, a few clarifications and cautions are in order.

First, writing about a social environment from within that environment is a uniquely challenging endeavor. As Marshall McLuhan and Quentin Fiore observed in *War and Peace in the Global Village*, "One thing about which fish know exactly nothing is water, since they have no anti-environment which would enable them to perceive the element they live in" [5] (p. 175). Our situation is not quite as hapless as fish, however, as we do have anti-environments with which to compare our digital world, namely primary orality and literate culture, but also analog electronic culture. Additionally, I believe we have occupied this ecosystem long enough that its effects are coming into clearer focus.

Second, while I am interested primarily in how digital computing and the environment it created structures consciousness, that is, in how our environment molds and shapes our patterns of thought, I recognize that the effects of the digital revolution are not limited to habits of mind. Media ecology, and especially the Toronto School of Communication, as reflected in the work of Harold Innis and Marshall McLuhan, pursued a broad range of media effects that, in addition to "cognitive consequences", included economic, social, and cultural change, as well as influences on personality [6]. Disentangling the various effects of digital computing on our world is likely not possible and perhaps not even desirable. Nonetheless, my primary, if not exclusive, focus will be on habits of mind.

Third, I want to acknowledge that this essay extends the work of other writers concerned with the ways our digital world is altering how we think and interact, including Nicholas Carr's *The Shallows: What the Internet is Doing to Our Brains* [7], Siva Vaidhyanathan's *Anti-Social Media: How Facebook Disconnects Us and Undermines Democracy* [8], and Jaron Lanier's *Ten Arguments for Deleting Your Social Media Accounts Right Now* [9]. But, whereas these projects are concerned with the platforms (Facebook, Twitter, TikTok, Instagram, etc.), applications (search engines, indexing sites, etc.), and infrastructure (the "Net") enabled by digital technology, the current project focuses on the effects of the underlying technology itself (i.e., microprocessing and digital computing) and how it alters sense-making. I do not intend to suggest that such digital "extensions" do not matter, only that there is value in beginning with the question: what are the effects of digital computing on our habits of mind?

It is important to address that question because the deep structural logics of communication technologies do not easily or readily avail themselves to conscious awareness, which makes their effects difficult to see and their influence difficult to resist. It is my contention that media forms function to train our habits of mind in much the same manner as do ideologies or archetypes: by repeatedly exposing us to unconscious patterns. When users engage with digital programs, platforms, and interfaces, they are necessarily immersed in digitality whose form, like the spoken and written word, has the capacity to structure thought [3]. It is vital, therefore, that we chart the deep structures of digital technology and how it unconsciously shapes our processes of sense-making.

Toward that end, I argue that digital computing and microprocessing cultivate a digital mind characterized by intransigence, impertinence, and impulsivity. In support of this argument, the essay unfolds in four stages. First, I reflect on a media ecology approach to communication technologies, drawing attention to Marshall McLuhan's notion of formal cause and its exploration through the tetrad. Second, I turn to digital computing, briefly reviewing its history and development as a means of highlighting the key components and characteristics of the microprocessor. Third, I analyze how efforts to mimic the "logic" of the microprocessor produces the digital mind. Finally, I discuss the implications and extensions of the preceding exploration, paying particular attention to its influence on our politics.

## 2. On Media Ecology

Media ecology approaches the study of media and communication from the premise that communication technologies—rather than simply being tools *in* our environment—*constitute* the very environment we inhabit. This approach further assumes that our social environment, as a product of our prevailing communication technologies, structures the form of social organization as well as human consciousness [11]. Media forms, notes Lance Strate, "*create the conditions that in turn condition us.* The changes that we introduce into our environment, that alter our environment, feed back into ourselves as we are influenced, affected, and shaped by that environment" [12] (p. 63).

For media ecologists, then, the history of society can meaningfully be organized according to the prevailing communication technologies at a given time, a perspective that generally results in the categorization of society into three major eras or epochs: primary orality, literate society, and the electronic age [13]. The literate world can further be divided into chirographic (writing) and typographic (print) cultures [10], and the electronic age grouped into analog and digital cultures.

The distinction between analog and digital culture is especially germane to the present study, as I am interested in how digital computing restructures consciousness. So, it is worth considering how they are distinct. Analog culture, as I am using it, emerges from electronic communication technologies such as telegraphy, wired telephony, cinema, broadcast radio and television, and mechanical computers that carry information using a "continuous and seamless signal" [14]. Digital culture, by contrast, reflects "a class of devices capable of solving problems by processing information in discrete form . . . expressed in binary code—i.e., using only the two digits 0 and 1" [15] (n.p.). Most electronic analog technologies have been replaced today by digital ones.

Having established the contours of my object, I turn to the matter of how one investigates the effects of a communication technology once it has established itself as the prevailing social environment. To do so, I draw upon Marshall McLuhan's understanding of *formal cause*, which he identified as the basis of his "own approach to media" [16] (p. 130), as well as central to his laws of media as reflected in the tetrad. Since McLuhan's view of formal cause revises Aristotle's, it is helpful to begin there. In Book V of his *Metaphysics*, Aristotle identified four causes that "account for how something comes into being as well as what makes something the particular thing that it is" [17] (p. 280). The four causes are material, efficient, final, and formal.

According to Aristotle, the material cause refers to the actual materials that constitute a thing, "the bronze of a statue and the silver of a cup", while the efficient cause describes the "source of the first beginning of change" (i.e., the actors and actions that led to its creation). So, in the case of a painting, for instance, the material cause would entail the canvas and paint, while the efficient cause would include the artist and possibly the technique of painting employed. The final cause captures the "end" (i.e., the purpose for which the thing is created), which in the case of a painting might be pleasure, enjoyment, or self-expression. Finally, by formal cause, which is our central concern in this essay, Aristotle meant something akin to its "*pattern*; that is the essential formula and the essential classes which contain it" [18] (p. 211). As is likely evident, formal cause is the most challenging of the four causes to pin down; it is also the one with which McLuhan takes the most liberties.

In "re-inventing" the concept, McLuhan understood "formal cause not in the sense of the classification of forms, but to their operation upon us" [19] (p. 259). Elaborating on this view, Thomas Sutherland notes McLuhan associated formal cause "with the effect that a medium has upon its audience" [20] (p. 254). Indeed, in *Media and Formal Cause*, Eric McLuhan quotes a letter his father Marshall had sent to a friend, Archie Malloch, in which he says, "formal causality is always [related to] the audience" [16] (p. 10). To appreciate how formal causality is related to the effects of a medium on an audience, it is helpful to consider Kenneth Burke's treatment of form [21].

In *Counter-Statement*, Burke describes form as the "psychology of the *audience*. Or, seen from another angle, form is the creation of an appetite in the mind of the auditor, and the

adequate satisfaction of that appetite" [22] (p. 31). While Burke's view of form concerned literature and the way "one part of a [work] leads a reader to anticipate another part" (p. 124), McLuhan's view of media technologies as *forms*, each with its own distinctive structural features, was very much consistent with Burke's view of literary forms. Essentially, McLuhan suggested that media forms implied a "*process* of change" by creating an appetite for the structural features of a given media form (medium), though he attributed this insight to Harold Innis [23] (p. 50).

In short, McLuhan maintained that every media technology has identifiable and relatively fixed structural features or traits [24]. Those features reflect the underling "logic" of that media form. Habitual use of a media form creates an appetite for that logic. When a media form comes to dominate our social and cultural environment, so too does the underlying logic of that media form. The process of change implied by a media form, i.e., the creation of an appetite for its structural logic and the adequate satisfaction of that appetite, which is achieved through the spread of its logic, is its effect. Formal causality is the approach that allows one to study the effects of media forms.

Toward the end of his life, McLuhan collaborated with his son Eric to formalize his approach into a "proper and systematic procedure" that they termed the laws of media [25] (p. 7). Their approach was based on "verifiable (that is, testable) statements" made in response to four prompts:

- What does the artefact enhance or intensify or make possible or accelerate? [...]
- What is pushed aside or obsolesced by the new "organ"?
- What recurrence or retrieval of earlier actions and services is brought into play simultaneously by the new form? [...]
- When pushed to the limits of its potential [...], the new form will tend to reverse what had been its original characteristics. What is the reversal potential of the new form? [25] (pp. 98–99)

These prompts, known as the tetrad, are frequently simplified into a single four-part question: what does a given media form enhance or intensify, obsolesce or displace, retrieve, and reverse into? In short, what does a media form want? I say "want" rather than "do" because the latter is a bit too deterministic, at least as far as formal cause is concerned. Just because a medium is structured to create a series of appetites does not guarantee that users will necessarily act on those appetites [4]. Alternatively, even though some users may take up a technology in unexpected (and uninvited) ways does not mean that that technology does not create an identifiable set of "structured invitations" or appetites. My aim, consistent with McLuhan's, is to understand the "formal" effects caused by the defining structural features of a media form, in this case digital computers and microprocessors.

I acknowledge that this approach—this kind of thinking—runs counter to how communication scholars typically treat "effects" today, namely as exclusively a product of efficient cause. The "obliteration" of concern with formal cause in favor of efficient cause is itself, according to McLuhan, an effect of the rise of print media and the culture of literacy [19], (p. 259). So, the relatively recent renewed interest in formal causality may be due, at least in part, to a weakening of print's rigidly linear logic. Before undertaking such an exploration, two additional points of clarification are in order. In invoking McLuhan's percept of formal cause, I potentially invite a series of associations with which I am somewhat less comfortable.

First, McLuhan famously suggested media are extensions of the human senses, which stresses the agency of users in relation to technology. In my view, it is more accurate to suggest that technologies extend their logics through human use. As Robert Logan has observed, "At first, technology serves as an extension of humankind, but after a while a subliminal flip takes place and suddenly the user is transformed into an extension of the technology they have come to consider part of them" [26] (pp. 430–431). McLuhan comes close to explicitly saying this in *Understanding Media* when he writes, "To behold, use or perceive any extension of ourselves in technological forms is necessarily to embrace it. ... By continually embracing technologies, we relate ourselves to them as servomechanisms" [27] (p. 46).

Second, I have deliberately avoided invoking McLuhan's most well-known aphorism, the "medium is the message", given its unfortunate suggestion—intended or not—that content does not matter to communication. It does [28], though admittedly content has more to do with *what* people think than with *how* people think. I am, however, comfortable with McLuhan's revised metaphor, expressed as "the medium is the massage". This phrase derives from the title of McLuhan's 1967 collaboration with graphic designer Jerome Fiore, *The Medium is the Massage: An Inventory of Effects*. The book's unlikely title, which was originally supposed to read the "medium is the message" arose from a typesetting error, but McLuhan liked the mistake and decided to keep it, writing in the book: "All media work us over completely. They are so pervasive in the personal, political, economic, aesthetic, psychological, moral, ethical, and social consequences that they leave no part of us untouched, unaffected, unaltered. The medium is the massage" [29] (p. 26).

To my mind, "the medium is the massage" is consistent with the view that technologies use us and not the other way around. In suggesting that media "work us over completely", this phrase underscores that technologies extend their structural logics into various spheres of life through human use. Importantly, the phrase does so without also implying that content does not matter. With these clarifications in place, I turn now to the defining traits or characteristics of digital computers and how they "massage" us.

## 3. The Structural Logic of Digital Computers

The central aim of this section is to identify and unpack the chief structural characteristics and, thus, logics of digital computers and microprocessors as a way to begin assessing the model of sense-making they foster. Toward that end, I offer a brief history of digital computers, identify the key components of microprocessors, distill the structural characteristics of microprocessors, and interpret those characteristics through the lens of the tetrad. In a subsequent section, I ask what happens when the human mind tries to mimic the binary, closed, and machine logics of digital computers and microprocessors.

Digital computers dominate virtually all aspects of our lives today. As Professor of Computer Science and Engineering, Arlindo Oliveira observes, "Computing technologies, which are only a few decades old, have changed so many things in our daily lives that civilization as we know it would not be possible without computers" [30] (p. 8). Despite the centrality and significance of computers in our lives, many people are unfamiliar both with their history and how they work.

Histories of computing often begin with the English mathematician Charles Babbage and the invention of the first *mechanical* computer, The Babbage Difference Engine, in the 1820s. A difference engine is a mechanical counting machine comprised of wheels, gears, and cranks that can perform the mathematical operations of addition, subtraction, and multiplication among other polynomial functions. It was a precursor to a more advanced counting-wheel machine proposed by Babbage, known as the Analytical Engine, though he never lived to see it built.

Nonetheless, Babbage's ideas, along with the development of Boolean logic by the logician and mathematician George Boole, made possible the creation of the Atanasoff–Berry computer (ABC), which is regarded as "the first electronic digital computer" [15] (n.p.). American physicist John V. Atanasoff and his graduate assistant, Clifford E. Berry, developed the ABC at Iowa State College in 1942. While the ABC reflected several innovations, the most important one was "using binary digits, just ones and zeros, to represent all numbers and data . . . [an innovation that] allowed the calculations to be performed using electronics" [31] (n.p.).

Decades later, computers continue to operate in this same basic manner, though their operations are performed by the computer's microprocessor. A microprocessor or central processing unit (CPU) is an integrated circuit or fabricated semiconductor chip that operates as the computer's computation engine. Typically, it is comprised of four main components: (1) an arithmetic and logic unit (ALU) that performs various mathematical calculations and Boolean functions; (2) a control unit that provides instructions and generates the signals to operate

other components; (3) a register array that temporarily holds data; and (4) input–output devices (I/O units) that transfer data between computers and external devices.

Based on these components and the functions they perform, digital computers and microprocessors are governed by three primary structural characteristics or logics: digitization, algorithmic execution, and efficiency.

### 3.1. Digitization

Digitization describes the conversion of all information and data into discrete numerical units, those units being either a 1 or a 0 [32,33]. While multiple authors have commented upon digitization, Walter Ong succinctly captures the centrality of digitization to computing technology:

> Ultimately, digitization today culminates in the computer, with its common binary digitization, the reduction of knowledge (data) to the most basic, most stark, most simple of numbers, binary digits, 0 and 1, which a computer 'handles' not by conceptualization (as human beings typically do) but simply by local motion, such as spatially separate units allow. Today, by digitization we commonly imply the use of complex instruments or technological contrivances to process (move around) vast, oftentimes billions, of those elemental units, counters, 0 and 1, 'no' or 'yes' Each 'bit' of information in a digital computer is always the result of a choice between two alternatives, 0 or 1, no or yes. Such 'bits' of information a computer moves around with amazing speed and complexity (often in clusters, or 'bytes'). But in the last analysis, moving around or manipulating such information—0 or 1, no or yes—is all a computer can do [34] (p. 71).

By dividing the world, or perhaps more accurately, our experience of the world, into discrete units, digitization transforms matter, which is made up of atoms, into "bits," and bits are "programmable, alterable and subject to algorithmic manipulation" [35] (p. 18).

### 3.2. Algorithmic Execution

Algorithmic execution refers to the fact that microprocessors are programmed to carry out a precise list of instructions or computations in a fixed and predetermined (step-by-step) sequence. As Theodore Roszak explains, a defining feature of computers is "the ability to process . . . information in obedience to strict logical procedures" [36] (p. xiv). Elaborating further, Oliveira writes, "Computers are useful only because they execute programs, which are nothing more than the implementation of algorithms. . . . Without algorithms, computers would be useless" [30] (p. 6). M Beatrice Fazi explains the logic governing algorithmic execution in *Contingent Computation*:

> [Digital computers are] *formal axiomatic systems*. An axiom is a self-evident truth; it is a postulate from which consequences are inferentially derived. Computational systems are axiomatic because they too implicate a starting point that is self-evident and which defines the inferential steps that can be taken from it toward s a result. From this perspective, computer programs, just like axioms, return to us only what is already known. They provide an output that is already implicit in the input [37] (p. 4).

Simply stated, computers are programmed to perform a series of specified tasks using specified data. While computers can do no more or less than what they are programmed to do, their programming has admittedly become considerably more complex in the past few decades.

### 3.3. Efficiency

Finally, efficiency means that microprocessors are engineered to perform as many calculations as rapidly as possible. Indeed, the desire to enhance processing power, to create microchips that can manipulate more information more quickly, has been the driving force behind the development of computing technology. In 1965, Gordon Moore, the co-founder of Intel, predicted the number of transistors that could be fitted on a silicon chip and,

hence, the processing power of computers would double roughly every two years. Nearly 60 years later, Moore's law has proved reliable, though experts agree computers will reach the physical limits of Moore's Law during the 2020s [38].

At the most basic level, digital computers and microprocessors are technologies designed to (1) break down information into discrete, binary units, (2) perform computations using those data in strict accordance with their established programming, and (3) to execute a large volume of computations with astonishing speed.

Seen through the lens of the tetrad, digital computers and microprocessors *enhance* the speed and capacity of computation (efficiency) through programmed responses (execution) to discrete data (digitization); *obsolesce* electronic analog media that, alternatively, are characterized by continuous data (contiguity), postmodern fragmentation (self-conscious perspectivism), and niche messaging (deliberateness); *retrieve* older technologies of precise calculation such as the abacus and slide rule; and *revert* into confusion and imprecision by producing more data more quickly than can be humanly comprehended. Figure 1 presents a tetrad of digital computers using the form proposed by McLuhan and McLuhan based on these fundamental properties [25] [5].

| | |
|---|---|
| **II. speed and capacity of computation through programmed responses to discrete data** | **I. confusion, imprecision** |
| **III. abacus, slide rule** | **IV. electronic analog media (contiguity, perspectivism, deliberateness)** |

**Figure 1.** Tetrad for microprocessor/digital computer.

## 4. Habits of Mind in a Digital World

Media and popular culture regularly compare human brains with computers, an analogy that, as Alan Jasanoff observes, contributes to the "tendency for people to view the brain as an abiotic machine" rather than as "biologically grounded and integrated into our bodies and environments" [39] (pp. 4, 7). One key figure to perpetuate the misguided isomorphism between brains and computers was the mathematician Claude Shannon, whose 1948 book, *The Mathematical Theory of Communication*, played a vital role in shaping early understandings of communication as involving encoded "signals".

But, brains and computers are fundamentally dissimilar, as suggested by their divergent histories. As Oliveira explains, "Computers are electronic devices, designed by humans to simplify and improve their lives; cells, the basic elements of all living beings, are biological entities crafted by evolution. ... Computers represent the latest way to process information, in digital form. Before then, information processing was done by living organisms" [30] (p. 1). Because of this dissimilarity, it is crucial that we explore how repeated use of digital computers and microprocessors habituate or condition the human mind. Based on the properties of digitization, execution, and efficiency, I contend that digital computers and microprocessors contribute to a *digital mind* characterized by intransigence, impertinence, and impulsivity.

Recognizing that the presentational form of information "massages" our understanding of that information, Table 1 presents the basic computer logics and their corresponding habits of mind in a tabular manner. This presentation is intended to augment the preceding linear discussion characteristic of academic writing.

**Table 1.** Computer logics and their corresponding habits of mind.

| Computer Logic | Habit of Mind |
| --- | --- |
| Digitization (binary code) | Intransigence: dichotomous and dogmatic |
| Algorithmic (input/output model) | Impertinence: close-minded and insensitive |
| Efficiency (machine logic) | Impulsivity: rash and hyper-affective |

*4.1. Intransigence: The Divisive, Dogmatic Mind*

In *Interface Culture*, Steven Johnson writes, "A computer thinks—if thinking is the right word for it—in tiny pulses of electricity, representing either an 'on' or an 'off' state, a zero or a one. Humans think in words, concepts, images, sounds, associations" [40] (p. 14). Johnson is right to be skeptical of using the word "think" when discussing computers. Unlike humans, computers do not think, they "compute", a process that begins with digitization. In this section, I suggest that repeated exposure to digitization fosters intransigent thought. By intransigent, I mean thought that is dichotomous and dogmatic.

"The world, as we experience it," explains Nicholas Negroponte, "is a very analog place. From a macroscopic point of view, it is not digital at all but continuous. Nothing goes suddenly on or off, turns from black to white, or changes from one state to another without going through a transition" [33] (p. 15). Since computers operate differently from human perception, constant exposure to digitization unconsciously urges us to divide our otherwise contiguous world into discrete units, units that are not only separate but also fundamentally opposed. In binary code, one and zero are opposing states; when the human mind tries to make sense of our analog world in this manner, it contributes to dichotomous, absolutist, and inflexible thinking [41].

This effect can be seen with respect to virtually any issue of great social importance, which persons increasingly align themselves "for" or "against". I wish to stress that I am not making an argument about *what* gets talked about today (content), but about *how* it gets talked about (form). In our digital world, most things—even facts and science—are treated as matters for debate, and "debate" is inherently an agonistic activity. Unlike discussion or deliberation, debate eschews collaboration, compromise, and common ground in favor of opposition, inflexibility, and self-interest. It pits people, groups, and even nations against one another, and indeed our digital world has witnessed a global rise in nationalism.

To more fully appreciate the habits of mind urged by digitization, it is helpful to construct a tetrad that describes the binary logic of digital computers. In Figure 2, I suggest that digitization *enhances* either/or logic, opposition, rigidity, and certainty; *obsolesces* both/and thinking, compromise, flexibility, and conditionality; *retrieves* agonistics; and *reverts* into cooperation, whose penultimate form is war.

| | |
| --- | --- |
| **II. either/or logic, opposition, rigidity, certainty** | **I. cooperation (in the form of war)** |
| **III. agonistics** | **IV. both/and thinking, compromise, flexibility, conditionality** |

**Figure 2.** Tetrad for digitization (binary logic).

Since this tetrad concerns a logic, quadrant III identifies the logic retrieved by binary code. In primary orality, agonistic procedures grounded in conflict and polemics prevailed. As Walter Ong explains, "Oral modes of storing and retrieving knowledge have much in common in all cultures. They are formulaic by design and, particularly in public life, tend

to be agonistic in operation" [42] (p. 123). So, while agonistics is not new, computers put a new spin on this logic. Rather than using polemics to arrive at more considered decisions, contemporary habits of mind favor dichotomous and dogmatic thinking. This is due, in part, to the shorter form of digitality (Twitter, for instance), as opposed to the longer form of orality (oral epics, for instance). While long form encourages complexity in debate, short form stresses simplicity, which entails the reduction of complicated issues into simple, often opposing ideas. As issues are stripped of nuance, people gravitate toward the emerging poles (either "for" or "against"), reducing the middle space that is vital for common ground and potentially compromise.

This intransigence of thought raises the specter of war by reverting into "cooperation", as suggested in quadrant I of the preceding tetrad. As Kenneth Burke has suggested, war reflects the ultimate act of cooperation because the parties involved have mutually agreed to stop searching for common ground (identification) and to battle one another until there is a victor [43]. Similarly, binary code allows for no middle ground; it is quite literally "uncompromising"—1/0, on/off, yes/no; it presents two discrete entities, whose meanings derive entirely through what they are not.

### 4.2. Impertinence: The Insensitive Mind

Unlike computers, which can do no more or less than what their programming allows, human beings are fully embodied, biological creatures, who continuously receive and respond to stimuli through the human sensorium. Lewis Mumford captures this distinction well when he writes, "computers cannot invent new symbols or conceive new ideas not already outlined in the very setting up of their programs. . . . [The human], on the contrary, is constitutionally an open system, reacting to another open system, that of nature" [44] (p. 191).

Moreover, as self-aware beings, humans have a capacity for reflection and morality that computers do not. So, what happens when the human mind tries to "process" information like a computer? If digitization and the logic of binary code foster an intransigent mind, one that is both dichotomous and dogmatic, algorithmic execution within a closed system habituates an impertinent mind, one that is increasingly insensitive and unresponsive to endlessly evolving contexts.

At a structural level, computers employ an input/output model rooted in if/then logic. While that logic may create the appearance of an open system because of its complexity (powerful algorithms with access to big data) and contingency (the possibility of multiple outcomes depending upon what data is input), it is not. Computers are bound by their programming and inputs. They do not make "choices," especially ethical or moral ones, and they do not experience doubt or regret. They execute commands in strict accordance with their programming, which is sometimes designed to simulate human qualities. But, the digital computer cannot, under any circumstance, as Mumford observes, "dream of a different mode of organization other than its own" [44] (p. 191).

Thus, computing is a closed—if admittedly complex and contingent—system, and when humans make sense of the world using this logic, they are less self-aware and sensitive. Again, our concern here is with form not content. Take matters of diversity and inclusion as a case in point. While there is certainly more talk about diversity today than in the past (content), the character of that debate (form) is not only highly polarized but also intensely caustic (think: trans rights or critical race theory). Persons on both sides are increasingly unwilling to seriously entertain alternative points of view, a fact that contributes to doxing, cancel culture, and other forms of violence.

To understand the habits of mind urged by algorithmic execution, Figure 3 presents a tetrad that highlights the closed system of digital computers. Specifically, I suggest that computer programming *enhances* an input/output model, if/then logic, and pre-established boundaries; *obsolesces* critical (reflexive) thinking, continuous feedback, and context responsiveness; *retrieves* the transmission model of communication; and *reverts* into (mis)interpretation, misunderstanding, and mistrust by incorrectly treating communication as transparent.

| II. input/output model, if/then logic, pre-set boundaries | I. (mis)interpretation, misunderstanding, mistrust |
|---|---|
| III. transmission model of communication (Shannon-Weaver) | IV. critical (reflexive) thinking, continuous feedback, responsiveness |

**Figure 3.** Tetrad for algorithmic execution (input/output).

Computing is "a closed system; language is an open system" [41] (p. 2). As such, the output of computers necessarily demands interpretation, which is the central argument of Walter Ong's *Language as Hermeneutic*. So, while computers retrieve the Shannon–Weaver model of communication, which formally heightens expectations for "perfectly transparent communication", their outputs, which are expressed through symbols and language, revert into misinterpretation and misunderstanding.

Ambiguity inheres in communication but not computers. The repetitive use of this media form, especially to interact with others, creates an appetite for untroubled communication, even as it reduces our ability to read subtle cues and microexpressions. "As the brain evolves and shifts its focus toward new technological skills," explain Small and Vorgan, "it drifts away from fundamental social skills, such as reading facial expressions during conversations or grasping the emotional context of a subtle gesture" [2] (p. 2). One consequence of the growing divide between expectation and reality is a lack of trust. "People deprived of interpersonal contact eventually suspect rather than trust others because their perception of reality has been skewed [by digital computers]," observes Michael Bugeja, "prompting misinterpretation of messages and motives, thereby harming relationships" [45] (p. 3).

### 4.3. Impulsivity: The Affective Mind

Computers, as previously noted, are engineered to be efficient, to execute as many commands and calculations as quickly as possible. They can do this effectively because they operate on electrical signals governed by binary code (discrete data) within a closed system (involving finite inputs). Furthermore, as machines, they produce objective, dispassionate, and near instantaneous results. Consequently, they can process data and render outputs at blazing speeds.

But, information processing and, ultimately, decision making in humans, especially reasoned, deliberative decision making, works differently. At its best, it is careful and considered, a slow, methodical, and reflexive process. Human concern with ethics and morality slows the processes of sense-making and action-taking even further. So, while computers are highly efficient at information processing, humans are comparatively less so. That difference matters because repeated exposure to the logic of efficiency formally invites humans to strive for greater efficiency. As humans attempt to mimic computer efficiency, they rely more heavily on instinct and affect. In short, as humans try to speed up their information processing and decision-making capabilities, they are less careful and rational and more impulsive and affective, which—paradoxically—undermines the quality of their decision making [6].

To understand the habits of mind urged by computer efficiency, Figure 4 presents a tetrad that describes the machine logic of digital computers. Specifically, I suggest that efficiency *enhances* the speed of information processing and decision making via immediate electrical signals; *obsolesces* reasoned deliberation, which is careful, considered, and consensus-oriented and, thus, inefficient; *retrieves* the mind/body dualism, which views the mind as abiotic; and *reverts* into embodied, affective response.

| II. speed of information processing and decision making | I. embodied, impulsive, affective response |
|---|---|
| III. mind/body dualism | IV. reasoned deliberation, consensus |

**Figure 4.** Tetrad for efficiency (machine logic).

As quadrant III suggests, the machine logic of computers rehabilitates the mind/body dualism by treating the computer as an extension of a disembodied mind. But, in privileging the principle of efficiency, digital computing reverts, through human use, into impulsivity and affective response. Commenting on this state of affairs, William Davies observes, "Knowledge becomes more valued for its speed and impact than for its cold objectivity, and emotive falsehood often travels faster than fact" [46] (p. xi).

An emphasis on the "speed of knowledge and decision making" does more than merely give way to impulsivity and emotionally charged responses though; reasoned deliberation and "consensus is sidelined in the process" [46] (p. xvi). In short, by inviting humans to make decisions impulsively, computers also necessarily invite humans to make them in isolation (individually). As such, computers contribute to a habit of mind that is not only emotion-driven, but also self-centered, which works to further affirm impertinence.

## 5. Implications and Extensions

In his 1985 book, *Amusing Ourselves to Death: Public Discourse in the Age of Show Business*, noted media ecologist Neil Postman lamented the state of public discourse in society at the time. Specifically, he traced how the transition from the age of typography (print) to the age of entertainment (television) caused an "epistemological shift" from a culture rooted in reason, seriousness, and exposition to one grounded in amusement, triviality, and image [47]. Since then, our culture has shifted again, this time from an age of analog electronic media (broadcast television) to an age of digital electronic media (computers and microprocessors).

Like previous shifts in prevailing technologies of communication, this one is accompanied by changing habits of thought. Whereas analog electronic media were defined by the structural properties of a continuous signal, postmodern fragmentation, and niche messaging, which reflect and promote the logics of contiguity, perspectivism, and deliberateness, digital electronic media are characterized by the structural logics of digitization (binary code), algorithmic execution (input/output), and efficiency (machine logic), which foster modes of thought characterized by intransigence, impertinence, and impulsivity.

In reflecting on this shift in communication technologies and, consequently, habits of mind, I wish to preliminarily probe three questions in this closing section: (1) What is at stake in such a shift? (2) What else matters to understanding the role of digital computing and microprocessing in changes to human consciousness and culture? And (3) what is next in terms of technological development and social change? Answering these queries will, given their scope, necessarily be somewhat more speculative than the preceding discussion.

### 5.1. What Is at Stake?

The central aim of this essay was to identify the underlying structural logics of digital computers and microprocessors and to highlight the mode of consciousness to which those logics give rise. Obviously, changing habits of thought have far-ranging consequences. So, it is worth reflecting on what some of those consequences likely are. The rise of the digital mind, I maintain, animates a series of broader cultural trends, including the deepening of political polarization, the spread of authoritarianism, and the decline of deliberative

democracy. In this section, I highlight a few of those possibilities, though it would require a book-length study to trace each of these connections and to properly explore how digital computing animates them.

While political polarization is certainly not new, recent data suggest that partisanship, as well as disdain for those who do not share our views, is spreading. According to a Pew Research Center study, "Growing shares in each party now describe those in the other party as more closed-minded, dishonest, immoral and unintelligent than other Americans. Perhaps the most striking change is the extent to which partisans view those in the opposing party as immoral" [48] (n.p.). In his 2021 book, *The Way Out: How to Overcome Toxic Polarization*, Peter Coleman notes that society is currently in the "grip of partisan contempt" [49] (p. 1), a grip that reflects an "extraordinary 50-plus-year pattern of escalation in political intolerance" [50] (n.p.).

Coleman does not point to a cause of our intensifying political polarization, but he does highlight twenty possible "macro drivers" such "governmental disjunction", "negative political campaigning", and "political gerrymandering" [50] (n.p.). I would note that these "drivers" are better classified as further symptoms of polarization than causes. Based on the analysis conducted in this essay, I want to suggest that the rise of digital computing and, consequently, the digital mind is a key, and perhaps even *the* key, contributing factor. After all, it combines a dichotomous and dogmatic pattern of thought with an emotionally charged insensitivity.

The same could be said of the global rise of authoritarianism and corresponding decline of deliberative democracy [51–53], both of which are flagged by William Davies in his 2018 book, *Nervous States: Democracy and the Decline of Reason*, as serious problems. According to Davies, there has been a fundamental shift in human thinking and decision making over the past four hundred years, a shift from rational deliberation aimed at consensus-building to a model that is combative, authoritarian, and emotive [46]. Like Coleman, Davies does not point to a cause. But, his historical accounting of this change corresponds to the technological shift from mass printing to digital computing.

Moreover, the development of digital computing has its roots in the military–industrial complex. As Davies notes, "Computers are originally instruments of war, as are the networks that connect them to each other. From the early 1940s through to the early 1960s, the needs of the US military drove the development of digital computers" [46] (p. 179). Because computers were engineered for the purpose of victory—for delivering "real time" knowledge—rather than consensus, they are not well suited to the "slow, reasonable, open debate of the sort that scientific progress has been built upon" [46] (p. 124). Democracy and deliberation are messy processes and, consequently, antithetical to the structural logic of computers.

*5.2. What Else?*

The question of "what else?" is inspired by McLuhan and McLuhan's recognition in *The Lost Tetrads of Marshall McLuhan* that "When you begin using tetrads to study media and artifacts, you find that each of the questions can produce more than a single answer". They continue, "All answers that are accurate are correct answers. When you have a clear answer to any one of the questions, then ask 'what else?' The results will often organize themselves into layers or rings" [54] (p. 7).

While digital computers constitute the basis of our social environment and, subsequently, act as filters through which virtually all other forms of communication pass, various media (plat)forms—each with their own unique structural characteristics—are built on top of that underlying architecture. As such, these digital platforms extend, modify, and potentially even undermine the structural logics of computers. Brian Ott has identified the defining structural logics of Twitter as simplicity, impulsivity, and incivility [55], while Vaidhyanathan has found that the structural features of Facebook include pleasure, surveillance, and attention [8]. But, it is vital that we continue to map the logic of these platforms.

Additionally, much of the recent scholarship on social media platforms investigates content concerns related to misinformation, disinformation, fake news, filter bubbles, and

echo chambers [56], as well the damaging psychological consequences of heavy social media use [57]. This work, which is typically social-scientific in orientation, establishes causal links between social media platforms and social and personal harms. While important, such work is ill-equipped to explain why those platforms produce those effects. It is crucial, then, that such work be augmented with scholarship from the perspective of media ecology, which in diagnosing *formal* causality aids us in making more informed, reflective, and productive choices about how we use technology and how it uses us.

Finally, digital computing is itself evolving. In addition to global networking, computers are increasingly defined by their mobility, portability, and wearability. The global networking of digital computers and the devices we use to interact and interface with networks all matter. One excellent model of this kind of scholarship is Dennis Cali's work on the logic of the link [58]. The interconnected nature of media today makes the practice of linking especially important, as does the interactive quality of media and the fact that users are always connected. But, we should be careful not to confuse hyperlinking, interactivity, and connectivity with community or citizenship, which experts suggests that digital technologies consistently undermine [59].

*5.3. What Next?*

In *Counter-Statement*, Kenneth Burke observed, "the conventional forms demanded by one age are as resolutely shunned by another" [22] (p. 139). In this context, "conventional forms" refers to literary genres, but as go literary genres, so go media forms. Basically, the technologies that dominate one era are replaced in another. Computers supplanted broadcast television, which replaced print, which displaced primary orality. Simply put, technology changes, and, in fact, the next momentous change is already underway. But, before identifying that change and speculating about it, some reflection on the pace of change is warranted.

Media ecologists and others who study technological change agree that rate of change is accelerating. "Since the dawn of [hu]mankind," writes Oliveira:

> cultures and civilizations have developed many different technologies that have changed profoundly the way people live. At the time of its introduction, each technology has changed the lives of individuals, tribes, and whole populations. Change didn't occur at a constant pace, though. The speed at which new technologies have been developed and used has been on the increase ever since the first innovations were introduced, hundreds of thousands of years ago [30] (p. 11).

So, it is perhaps not surprising that, even though the shift from analog electronic media to digital electronic media is, roughly, only 30 years old, that we now find ourselves on the precipice of another great shift, one ushered in by the emergence of artificial intelligence (AI). Like previous technological revolutions, AI will restructure human consciousness and social organization yet again. While extended analysis of the structural properties and, hence, logics of AI is beyond the scope of this essay, I will point to the general trajectory of technological development by humans in which it participates.

Initially, the science of artificial intelligence openly sought to create machines that could mimic the "natural intelligence" of animals, especially humans, but modern AI claims to have abandoned that quest in favor of computational intelligence, which according to Poole, Mackworth, and Goebel, "is a system that acts . . . appropriate[ly] for its circumstances and its goal, [. . . ] is flexible to changing environments and changing goals, [and] learns from experience" [60] (p. 1). Even this brief one-sentence introduction to AI hints at the ways in which it differs from digital computers, which are not particularly context sensitive, adaptive to changing conditions, and capable of learning from experience.

The science of AI is animated by a desire to endow machines with a series of capabilities that include, but are not limited to, reasoning, planning, knowledge representation, machine learning, perception, and affective intelligence. This list betrays the claim of AI scientists who insist that they are no longer trying to mimic the human brain. But, as we have seen throughout this essay, machines are not biological entities. So, the only way to

overcome that disparity is to begin to create and build machines that are fully compatible with human biology, which is exactly the direction that AI research is moving [61].

All of this is by way of saying that artificial intelligence is not concerned with building super powerful computers; it is concerned with reengineering machines to integrate them with human biology. Contrary to dystopian depictions of the future in which self-aware machines rise up and overthrow their human creators, humans are becoming the machines. The coming AI revolution, which is hundreds of years in the making, will not extend the human senses, it will rewire them in ways that fundamentally alter what it means to be human.

While it is tempting to simply conclude this essay on the preceding thought, it strikes me as bit too stark, too settled, too—in a word—computer-like. I have authored this essay not out of a sense of resignation but out of a sense of something more human: hope. I am hopeful that in drawing attention to our relationship with technology and to the centuries-old trajectory of that relationship, one in which, to borrow a phrase from Kenneth Burke, humans are increasingly *"separated from [their] natural condition by instruments of [their] own making"* [28] (p. 13), that we will make thoughtful choices about the instruments, especially the communication technologies, we create and whom, in turn, they invite us to become.

**Funding:** This research received no external funding.

**Institutional Review Board Statement:** Not applicable.

**Informed Consent Statement:** Not applicable.

**Data Availability Statement:** Not applicable.

**Acknowledgments:** The author wishes to thank Gordana Lazić and Jay Howard for their willingness to discuss the ideas and arguments reflected in this manuscript.

**Conflicts of Interest:** The author declares no conflict of interest.

## Notes

1.  On the nature and importance of the distinction between brain and mind, see Lewis Mumford. The mind is a "durable mode of symbolic organization . . . superimposed upon [the] purely electro-chemical changes . . . of the brain itself, a private organ". Elaborating further, Mumford notes that the mind creates "a sharable public world of organized sense impressions and supersensible meanings: and eventually a coherent domain of significance" [4] (pp. 26–27).

2.  I acknowledge that my use of the term "consciousness" is somewhat idiosyncratic. Rather than using it in a generic sense to refer to self-awareness, I mean something more like habits of mind or patterns of thought. I take consciousness to refer to our ways of thinking and our modes of information processing and sense-making.

3.  In *Orality and Literacy*, Walter Ong demonstrated that the spoken and written word are fundamentally different, producing distinctive environments and cultures that, in turn, condition human thought and behavior differently [10]. This essay concerns how the digital *word* and digital *world* are every bit as distinctive from the literate world as primary orality is from literacy. Expression and, by extension, thought vary greatly depending upon the medium of communication and its deep structural logics. Importantly, the consequences of those differences do not depend upon one's conscious awareness of them. It matters a great deal, for instance, that the preceding paragraph, as well as this entire essay, was composed on a computer rather than delivered orally, even though both involve language. The digital composition erases the materiality of my voice, its pitch, timbre, and tempo, transforming the word(s) from an event into an object. More difficult to perceive, though no less important, is how a digital composition differs from a written one. The potential for a swift and very public reply to a digital composition, as well as its circulation in an entirely different information environment, alter the form of expression at every stage from creation to consumption. Typing (to say nothing of texting) and writing are as different as are writing and printing. Similarly, the end product digital "text" is fundamentally different than the printed page. And, finally, scrolling on a screen and turning the page of a book are different sensory experiences. These differences all matter and produce different psychodynamics.

4.  This is the matter of free will. Media forms, like rhetoric generally, create "structured invitations". They are organized or structured to elicit particular kinds of responses; they urge users/audiences to engage in specific practices of sense (meaning) and sensibility (thinking and feeling) making. But, they cannot guarantee them, as users and audiences may act and respond in novel ways. I wish to stress that the deep structural logics of technologies "cultivate" particular habits of mind, but they do not irresistably impose them. Unlike computers, which can do no more than their programming allows, humans have a unique capacity to think and act in unpredictable ways. This is tied, at least in part, to the symbolic nature of human communication and its strong potential for misinterpretation and misunderstanding.

5    My tetrad for digital computers shares similarities to the one McLuhan and McLuhan arrived at in *Laws of Media*. They suggest computers enhance speeds of calculation and retrieval; obsolesce sequence, approximation, perception, and the present; retrieve perfect memory—total and exact; and revert into anarchy via the overlay of bureaucracy [24]. The most significant departure, I believe, lies in quadrant III, in which they suggest computers retrieve perfect memory. But, memory has never been perfect or total. So, I am not sure how computers could retrieve it. To my mind, it is more useful to suggest computers retrieve the mathematical clarity and precision of older technologies like the abacus.

6    At this point, I wish to acknowledge that there are absolutely contexts in which human decision making is significantly aided by the use of computers. Investing is a good example, and investment advisors are almost certainly better at their jobs as a result of using computers that can compare millions of data points over a specified time interval very rapidly. No human being could do this. But humans using computers as tools to accomplish specific tasks is very different than my central concern in this essay, which is how repeated exposure to computing alters human sense-making. The ubiquity of digital computing trains our consciousness and invites us to develop a digital mindset regardless of context.

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
