# Peer review of "The Digital Mind: How Computers (Re)Structure Human Consciousness"

_philosophies, doi:10.3390/philosophies8010004_

Round 1

Reviewer 1 Report

This study is a philosophical interpretation of the features of the meaning-making of people in the era of the development of technological communications. The topic of the article is relevant. The sources that the author draws are necessary for the disclosure of the topic. The abstract reflects the content of the work.

Author Response

I wish to thank Reviewer 1 for their generous and supportive feedback. The reviewer did not make any specific suggestions for revision. So, the revisions and edits I made to the manuscript will be addressed in my response to Reviewer 2.

Reviewer 2 Report

This is a very interesting and provocative article that raises a number of very important questions that will be of interest to a wide range of readers. The thesis is clearly stated and the topic is laid out in a cogent manner that wisely defines the terms clearly and then goes over some of the history of computing before using precise models to explain how digital computers may affect our ways of thinking.  One general issue I saw was that the theorists with whom the author engages are almost all from the 1960s which gives the theoretical apparatus a somewhat dated feel. There has been an enormous amount of work done in media studies since McLuhan and Ong and one would have liked to read more about what some of the more recent theorists have to say on this matter. However,  authors have the right to choose the writers with whom they wish to engage so inasmuch as the choice was made to analyze computers in terms first laid out by these scholars, I would not insist on bringing new voices into the mix, but would only suggest it in order to engage with the latest scholarship on this issue.

The main issue that I have is with some of the key points of the thesis itself that I will explain below, along with some comments on specific lines that could use revising or clarification.

Line 70ff:  My main objection is that I wonder if the structure of the underlying digital technology really has an effect on our ways of thinking, independent of the platforms that are used?  This is a central question in the article and I am not sure that I am convinced of this by the article. The article does make a very good case that technology affects our ways of thinking and the article does instill in the reader a lot of sympathy for the approach of McLuhan, media ecology and formal causation.  What I find problematic is the idea that the digital features of the programming code can affect our thinking, rather than just the actual user interfaces, programs, and platforms that humans engage with.  While the author does an excellent job ( though I disagree with some specifics as I will say below) of explaining the way that digital features such as binary code, Boolean operations and such can affect the way we think, the mechanism as to how we would be affected is unclear to me. How would these features affect us if the general public is not even aware of what they are or how they operate, as the author basically states in line 26.  These features lie deep in the code and are generally not exposed to human perception. I am not sure I see how the underlying code that powers a program, such as Microsoft Word for example, would have any affect one way or the other on a person using Word. Whether microprocessors operate in binary or not does not affect the infinite variety of things that can be written on the page by a human using Word. Maybe this is stretching it too far, but does the fact that paper is made out of trees instill any particular relationship between a writer and the forest?  This is actually very similar to a problem that has garnered a lot of attention in physics, namely the question of whether the strange quantum effects that occur at the microscopic level have any effect on the macroscopic world. Many scientifically minded philosophers of spirituality such as Fritjof Capra, Gary Zukav and Deepak Chopra have contended that the great insights of the world’s mystics can be justified by our new understanding of the quantum world, and that quantum effects may be responsible for such things as free will, ESP, Psychokinesis or a whole host of unexplained phenomena. But the vast majority of scientists say that what happens at these deep levels actually has no bearing on the regular material world in which we live. I think this issue should be addressed more clearly in the article, and the author should at least acknowledge this possible objection and explain a little more about how it might be countered.

 Second, regardless of the mechanism, I think more discussion is warranted about whether the effects are truly what the author claims they are, namely intransigence, impertinence and impulsivity.  I am not saying they aren’t, but again just like the comment above, some more discussion of possible objections would greatly strengthen the article. For example:

On line 357 it says that constant exposure to digitization teaches us to divide the world into discrete and oppositional units. But computers can and do handle fuzzy logic all the time, and can deal with different alternatives, and can change their output though machine learning and perform many operations that, while ultimately programmed in binary code, are not binary at all at the manifest level.  In fact the user generally doesn’t even know how the code itself operates, so why would they find their thinking bifurcated into yes/no intransigent terms?    

There are a whole host of counter examples that one could adduce for the idea that we are becoming more intransigent, for example many people are far more tolerant of difference than they were before the digital age, as shown by the fact that trans rights are a major concern of many people in America today, which I submit would not likely have developed without the social media platforms that have been used to introduce people to the ideas surrounding this movement, and to the very trans people themselves, whom people in smaller towns might never have come across otherwise. Likewise, whereas in the past people living in enemy states were easily vilified, currently there is much sympathy for, say Iranians because while there may be great conflict at the political level, one might be part of an online role playing game with Iranians and get to know them as human beings through that.  

Regarding impertinence, around line 393 in the discussion of Boolean logic, the author should account for the Or / Else functions which add an enormous range of possibilities and counter possibilities to what the author presents as a more fixed series. Not to mention that “if” also sets up the possibilities of dealing with different states that can alter and change over time or during the course of the implementation of the algorithm.

At line 438 where the author discusses the connection between the speed and efficiency of computation with impulsivity and relates it to a reversion back into more emotionally based decision making, it should also be noted that the speed with which computers operate is not just disembodied “speed” but speed OF something, namely information processing. In that sense, computers may make decisions more quickly, but they are not based on emotions but on actually having much more data upon which to base the decision at hand. A very good example of this is with investment decision making. Before the age of computers, no investment advisor could really compare millions of data points about prices over 20 years of one stock compared to another, but now they can, and can therefore make far better advised suggestions of what to invest in based on enormous amounts of previous data, rather than gut feeling which is what it came down to before computers.  It may be true that taking more time to make a decision is better than impulsive ones, but having more data upon which to base ones decisions seems to me even better than more time. Without more data, the time may be spent merely on mulling over and over the same ideas, and not really advancing knowledge of the situation.

Again, I do not wish to compel the author to change the thesis or conclusions, I am just presenting some counter arguments that I think the article would be well advised to consider.

Here follow some other unrelated points that I noticed in the article:

-          Some clearer explanation of the different causes and especially what Aristotle meant by “formal cause” around line 130 might be helpful to the reader.

-           In the tetrad figures starting with Figure 1 and going through all the figures, there should be a label on each quadrant because it is unclear which question each quadrant is addressing from the way it is laid out.

-          On line 331 the authors says that brains and computers are fundamentally dissimilar but cites a quote that does not discuss the operational dissimilarities but mostly the raw fact that one was designed by humans and the other by evolution, but this does not adequately explain their differences only their historical roots.

-          At 378 there should be some more explanation as to how war reflects the greatest degree of cooperation because the parties have stopped searching for middle ground. I am not sure I see the connection here between war and cooperation, unless the author means that within each side, the communities must cooperate mightily to fight the enemy perhaps in the way all elements of British society came together during WWII.

I I am also unclear about what citation format is being followed in this paper?  The references do not appear to be laid out correctly, as the page numbers are given in the text and in the references. 

Thank you for the opportunity to engage with this thought provoking paper

Author Response

I wish to thank Reviewer 2 for their careful and incisive feedback. The reviewer is clearly a well-versed content expert in this area, and I appreciate the opportunity to address their questions and concerns. Before I turn to an overview of my specific edits and revisions, which I marked in the manuscript using track changes, I also wish to thank Reviewer 2 for the kind tone and generous quality of their review. The feedback was helpful, encouraging, and supportive even when the reviewer was not fully compelled by my argument and analysis. I very much appreciate that the reviewer was oriented toward helping me to improve the essay I wrote rather than imposing their vision upon me. That approach has, I believe, resulted in a stronger manuscript.

In my reading, Reviewer 2 expressed two global concerns and made a series of more minor suggestions. As detailed below, I have tried in one manner or another to address each of these issues. The first broad concern raised by Reviewer 2 was that I had not adequately been clear about the mechanism or process by which the deep structural traits and logics of a technology might contribute to identifiable habits of mind. This comment insightfully cuts to the heart of the essay, and I have tried to resolve it in two ways. First, I added a new paragraph to the introduction just before the thesis statement that conveys my perspective and guiding assumptions on this matter. Specifically, I maintain that the process is akin to the function of ideology and, therefore, operates on a largely unconscious register. This approach to understanding habits of thought or patterns of mind is consistent with the philosophy of Walter Ong. So, second, I address Ong’s thoughts on this matter in an extensive new endnote (#3), drawing connections between his work and the present study.

The second global concern raised by Reviewer 2 was that I had not yet made a compelling enough case for intransigence, impertinence, and impulsivity as habits of the digital mind. This led to a substantive rewrite of each of those three sections. The new text is highlighted using track changes, though I deleted the old text in these areas so that my revisions would be clearer. Here is an overview of my changes:

(1) With respect to intransigence, I am much more explicit that this logic concerns not so much “what” we think or say on a specific subject but “how” we approach and talk about virtually all subjects today. Increasingly, I argue, virtually all things – including facts and science – are treated as matters for debate. I am less concerned with the specific contours of any give position on a subject (take COVID vaccines, for instance) than I am with the fact that the public “debate” rapidly devolves into the polarized positions of pro and anti-vaccination.

(2) With respect to impertinence, I jettisoned the argument about Boolean logic altogether and, indeed, I have removed most references to it throughout the manuscript. This was a poor descriptor of the habit of mind I was trying to get after, which really has more to do with the fact that computing, while complex and contingent, is a closed system. While I still think an analysis of Boolean logic is worthwhile, it was misplaced here. The revised analysis tracks more closely with my discussion of the digital logic of algorithmic execution. This revision allowed me to use diversity as a supporting example. I highlight, for instance, that public debates around issues like trans rights and critical race theory rapidly become not only polarized, but also caustic. Response to J. K. Rowling offers a particularly clear example of this.

(3) With respect to impulsivity, I am clearer in the revised manuscript that my central concern is with how the logic of efficiency is creating a habit of mind that effectively makes human decision-making more impulsive and affective. I clarify that this broad habit of mind is different from human use of computers as tools to accomplish specific tasks. There is no question that computers aid humans in doing things they could not otherwise do, but this essay concerns the unseen and unintended consequences of repeated use of a given technology. The example of investment advisors raised by the reviewer is a helpful one in this regard, and I added an endnote (#4) specifically addressed to this.

In addition to these more global concerns, the reviewer highlighted a series of places where the essay would benefit from minor edits. In response, I have (1) labeled the quadrants on all tetrads; (2) reframed the quotation used as support on line 331 of the original manuscript; (3) clarified how agonistics reverts into cooperation, whose ultimate (or “purest”) form is war (after all, you can’t really be at war with someone who refuses to do battle with you; you may have a hostile invasion and takeover, but not war). Finally, I would not that I concur that the reference system, which includes both notes AND page numbers, is odd, but as near as I can tell that is the requested style on the website and seen in the journal. If I’m mistaken about that, I’m happy to correct it.

I will close by highlighting one suggestion that I did not take, which involves the literature I cite. I would frame my reliance on “older” scholarship (like McLuhan and Ong, for instance) as a commitment – per my academic training – to primary sources over secondary sources (scholars writing about other scholars and their work). Basically, I rely on primary sources unless the specific interpretation of a primary source offered by a secondary source is directly relevant to the point I’m making. As such, where appropriate, I draw on well-known interpreters of McLuhan such as Strate, Logan, and Anton, and well-known interpreters of Ong such as van den Berg and Chesebro.

Again, my deepest gratitude to the reviewer for their helpful insights and suggestions throughout. I hope it has resulted in a more compelling version of the manuscript.